# Comparative Analysis of the Upper Respiratory Bacterial Communities of Pigs with or without Respiratory Clinical Signs: From Weaning to Finishing Phase

**DOI:** 10.3390/biology11081111

**Published:** 2022-07-26

**Authors:** Pabulo Henrique Rampelotto, Anne Caroline Ramos dos Santos, Ana Paula Muterle Varela, Karine Ludwig Takeuti, Márcia Regina Loiko, Fabiana Quoos Mayer, Paulo Michel Roehe

**Affiliations:** 1Laboratório Experimental de Hepatologia e Gastroenterologia, Centro de Pesquisa Experimental, Hospital de Clínicas de Porto Alegre, Porto Alegre 90035-007, Brazil; prampelotto@hcpa.edu.br; 2Laboratório de Virologia, Departamento de Microbiologia, Imunologia e Parasitologia, Instituto de Ciências Básica da Saúde, Universidade Federal do Rio Grande do Sul, Porto Alegre 90050-170, Brazil; anne_carolsantos@hotmail.com; 3Centro de Pesquisa em Saúde Animal, Instituto de Pesquisas Veterinárias Desidério Finamor (IPVDF), Departamento de Diagnóstico e Pesquisa Agropecuária, Secretaria de Agricultura, Pecuária e Desenvolvimento Rural, Eldorado do Sul 92990-000, Brazil; anapaulamut@gmail.com (A.P.M.V.); bimmayer@gmail.com (F.Q.M.); 4Campus III, Universidade Feevale, Novo Hamburgo 937000-000, Brazil; karinelt87@yahoo.com.br (K.L.T.); marcialoiko@gmail.com (M.R.L.); 5Departamento de Biologia Molecular e Biotecnologia, Instituto de Biociências, Universidade Federal do Rio Grande do Sul, Porto Alegre 90650-001, Brazil

**Keywords:** piglets, microbiota, next-generation sequencing, metataxonomy, metabarcoding

## Abstract

**Simple Summary:**

In this work, we performed a prospective study to compare bacterial communities in the nasal and laryngeal cavities of pigs with or without clinical signs of respiratory disease which were followed in a longitudinal fashion, at three critical phases of production, from weaning to the finishing phase. The findings reported here provide evidence that the composition of the upper respiratory tract bacterial microbiota differs significantly when comparing pigs with or without respiratory clinical signs after weaning; these differences were maintained in the nursery phase but were not observed at the finishing phase. Our results contribute to the knowledge of the porcine microbiota at different stages of production, providing new insights into the role of bacteria in the early stages of respiratory diseases.

**Abstract:**

A prospective study was conducted to identify bacterial communities in the nasal and laryngeal cavities of pigs with or without clinical signs of respiratory disease in a longitudinal fashion, from weaning to the finishing phase. Nasal and laryngeal swabs were collected from asymptomatic pigs (n = 30), as well as from pigs with clinical signs of respiratory disease (n = 30) at the end of the weaning (T1—33 days) phase, end of the nursery phase (T2—71 days), and finishing (T3—173 days). Total DNA was extracted from each sample, and the V4 hypervariable region of the *16S rRNA* gene was amplified and sequenced with the Illumina MiSeq platform. Principal coordinates analysis indicated no significant differences between the nasal and laryngeal bacterial communities. Nevertheless, the microbiota composition in the upper respiratory tract (URT) was clearly distinct between animals, with or without signs of respiratory disease, particularly at post-weaning and the end of nursery. In pigs with clinical signs of respiratory disease, *Actinobacillus*, *Streptococcus Porphyromonas*, *Veillonella*, and an unclassified genus of Pasteurellaceae were more abundant than in pigs with no signs. Metabolic prediction identified 28 differentially abundant pathways, mainly related to carbohydrate, energy, amino acid, anaerobic, and nucleotide metabolism in symptomatic pigs (especially in T2). These findings provide evidence that the composition of the URT bacterial microbiota differs significantly when comparing pigs with or without respiratory clinical signs after weaning, and this difference is maintained in the nursery phase; such differences, however, were not evident at the finishing phase.

## 1. Introduction

Swine respiratory diseases are multifactorial conditions that affect growing pigs in different stages of production, causing long-standing herd problems and significant economic losses. The etiology of respiratory problems in pigs is complex, usually involving a combination of infectious agents and environmental stressors that affect the pigs’ health and result in reduced feed efficiency and growth rate, increased treatment, and medication costs, as well as increased morbidity and mortality [1]. 

Confined populations in commercial farming and production systems are subjected to stressful environments that may reduce immunity, thus contributing to the occurrence of respiratory diseases. Environmental conditions, such as temperature, dust, ammonia, carbon dioxide, and intense farm management, influence the overall pathogen load, intensity and frequency of pathogen exposition, and pig immune system [2]. Disease outcome, in turn, depends on the balance between the pathogen pressure and the pig’s ability to cope with them.

Among the main infectious pathogens commonly related to respiratory disease, there are a variety of viral and bacterial agents, including influenza A virus (IAV), porcine reproductive and respiratory syndrome virus (PRRSV), *Mycoplasma hyopneumoniae*, *Mycoplasma hyorhinis*, *Pasteurella multocida*, *Glaesserella parasuis*, *Bordetella bronchiseptica*, *Actinobacillus pleuropneumoniae*, and *Streptococcus suis* [3]. The presence and abundance of these pathogens vary significantly among farms, production sites, regions, and countries, making infections difficult to treat and control. In addition, other microorganisms that reside in the upper respiratory tract of pigs may be involved in the development of respiratory diseases, thus highlighting the importance of a complete characterization of the swine’s nasal microbiome. Knowledge of the swine microbiota and its fluctuations may contribute not only to improving the quality of production but also to understanding the determinants of health or disease in pigs, as well as the other species that pigs may interact with, including humans [4].

Studies on the microbial diversity of pigs have mainly targeted the gut microbiome [5,6,7,8,9], while the microbiome of the upper respiratory tract (URT) has been less frequently investigated [10,11,12]. The URT is colonized by a multitude of microorganisms, most of which bear no association with disease [13]. Yet, such colonization will provide an initial line of defense against potentially pathogenic agents and occupy sites that could otherwise become targets for pathogens [10]. Microbial colonization may also act on host immunity, thus altering susceptibility to disease [14,15].

Previous studies on the pig URT microbiome have focused particularly on defined stages of commercial pig production, such as the first weeks of life [10] or finishing phase [16]. Therefore, sequential analyses of respiratory microbiota composition at critical points during pig production could provide insights into the evolution of microbial colonization throughout the production process. In the present study, a prospective study was carried out to compare the bacterial communities of the URT of asymptomatic and respiratory disease-affected pigs, which were followed in a longitudinal fashion at three critical phases of production. 

## 2. Materials and Methods

### 2.1. Herds

A nursery, which housed a total of 1900 pigs, was selected for the study. The farm was located in Rio Grande do Sul, Southern Brazil, which was selected based on previous reports of respiratory clinical signs. The pigs were housed on the nursery farm from 21 to 71 days of age, when they were moved and housed in a finishing facility until the day of slaughter (around 180 days of age). Thus, the first two samplings were performed on the nursery farm, with the last one on the finishing farm.

The following vaccination protocol was adopted at weaning (21 days of age) for the pigs of the study: *Actinobacillus pleuropneumoniae* (autogenous vaccine, Microvet), *Mycoplasma hyopneumoniae* (M+PAC, MSD Saúde Animal), and porcine circovirus type 2 (PCV2; Circumvent PCV, MSD Saúde Animal). Antibiotic treatment in feed was administered as part of the farm management (Appendix A). Tulathromycin was administered intramuscularly (Draxin^®^, Zoetis™), and norfloxacin (Farmaflox^®^, Farmabase™) was supplied in the drinking water when pigs presented respiratory clinical signs.

### 2.2. Study Design and Sampling

A total of 20 pigs were selected and sampled at three different time points, resulting in 60 samples. They were divided according to clinical status, such as asymptomatic (n = 10) and symptomatic (n = 10, with clinical symptoms of respiratory disease at T1). The selection criteria were based on the presence of coughing and sneezing, which were assessed and counted after movement stimulation, according to Morés et al. (2001) [17]. Three two-minute counts were performed, and the average of three counts was used to define the groups. Body temperature was measured using a digital rectal thermometer, and anamnesis was performed to monitor clinical signs. These animals were followed within three time points, representing different phases of pig production, i.e., post-weaning (T1—33 days), end of the nursery phase (T2—71 days), and finishing—prior to slaughter (T3—173 days).

Sterile nasal swabs were introduced in the nasal cavity by rotating them clockwise and counterclockwise, and sterile laryngeal swabs were obtained using a mouth gag and a laryngoscope for the introduction of the swabs in the larynx [18]. All samples were stored at −80 °C until further processing. The study took place from winter (post-weaning sampling) to spring (finishing phase prior to slaughter).

### 2.3. DNA Extraction, PCR, and Sequencing

Prior to DNA extraction, samples were homogenized in Precellys^®^ 24 (Bertin Technologies S.A.S, Versailles, France). Microbial DNA was extracted using the MagMax Pathogen DNA/RNA kit (Thermo Fisher Scientific, Waltham, MA, USA) at the Laboratório de Virologia, Universidade Federal do Rio Grande do Sul (Porto Alegre, Brazil), according to the manufacturer’s instructions. The V4 region of the bacterial *16S rRNA* gene was amplified by PCR (94 °C for 2 min, followed by 35 cycles at 94 °C for 30 s, 55 °C for 30 s, and 68 °C for 45 s, as well as a final extension at 68 °C for 5 min) using 16S—forward (5′ TCG TCG GCA GCG TCA GAT GTG TAT AAG AGA CAG GTG CCA GCM GCC GCG GTA A 3′) and reverse (5′ GTC TCG TGG GCT CGG AGT TGT GTA TAA GAG ACA GGG ACT ACH VGG GTW TCT AAT 3′) primers’, which contain the Illumina adapter sequence (Kozich et al., 2013). The reactions were performed with Phusion polymerase and 12.5 ng of DNA, following the standard protocol for this enzyme. Amplicons were purified with Agencourt^®^ AMPure^®^ XP beads (Beckman Coulter, Brea, CA, USA) and indexed following the Illumina protocol. Libraries were then quantified using Qubit 2.0 fluorometer (Invitrogen, Carlsbad, CA, USA), and concentrations were normalized for sequencing, which was performed with Miseq Reagent kit v2 500 cycles (2 × 250 paired-end) through the Miseq desktop sequencer platform (Illumina) located at the Instituto de Pesquisas Veterinárias Desidério Finamor (Eldorado do Sul, Brazil). Of the 36 samples, six could not be sequenced (S23, S28, S29, S30, S34, and S35) and, therefore, were not included in the study.

### 2.4. 16S rRNA Reads Processing

The sequence data exported from the MiSeq System was processed using a custom pipeline in Mothur [19]. Initially, sequences were depleted of barcodes and primers (where no mismatches were allowed). A quality filter was then applied to eliminate low-quality reads. Quality control was conducted by trimming the low-quality reads, including those with inadequate lengths, containing ambiguous bases, or with homopolymers longer than 8 bp. All potentially chimeric sequences were identified and removed using VSEARCH [20].

After the initial quality filtering and trimming steps, the remaining sequences were clustered into operational taxonomic units (OTUs), based on a 99% identity level, and classified against the SILVA v138 reference database at 97% similarity. Sequences that could not be classified (i.e., “unknown” sequences), as well as sequences identified as eukaryotes, mitochondria, and chloroplasts, were removed prior to further analysis.

An additional filtering step was performed by removing OTUs with less than 10 reads, in order to reduce spurious OTUs caused by PCR or sequencing errors. After all filtering steps, the resulting OTU table was composed of 1894.081 sequences, with an average of 75,763 sequences per sample (Appendix A). The OTU table was then rarefied to the smallest library size. Subsequent analyses of the sequence dataset were performed in R v. 4.0.0 (using vegan, phyloseq, ggplot2, and MicrobiomeAnalystR packages) or QIIME2 [21].

### 2.5. Microbial Communities and Statistical Analysis

Alpha diversity was assessed using species richness (Chao1) and diversity (Shannon and Simpson) indices. For overall comparison of significant differences among bacterial communities (i.e., beta diversity), principal coordinates analysis (PCoA) based on Bray–Curtis dissimilarity metric was performed. To achieve statistical confidence for the sample grouping observed by PCoA, a permutational multivariate analysis of variance (PERMANOVA) was performed on the distance matrix. To compare additional differences among the microbial communities, clustering methods based on Bray–Curtis dissimilarity were performed. The results of hierarchical clustering were visualized using heatmaps and dendrograms.

To detect potential taxa biomarkers, the linear discriminant effect size (LEfSe) method was performed [22]. The algorithm performs a nonparametric factorial Kruskal-Wallis sum rank test and LDA to determine statistically significant different features among taxa and estimates the effect size of the difference. Benjamini–Hochberg adjusted *p*-value was calculated to control the false discovery rate (FDR) in multiple tests. Differences were considered significant for a LDA score > 2.0 and FDR corrected *p*-value of 0.05.

### 2.6. Functional Prediction

Predictive functional gene profiling was based on 16S rRNA gene sequencing data using Piphillin with an updated KEGG database (from May 2020) and confidence cutoff value of 97 [23]. The resulting predicted metabolic pathways were then filtered to include only microbial ones. Dendrogram of KEGG pathways was calculated using the Bray–Curtis metric. Differentially abundant features were determined using LEfSe [22]. Benjamini–Hochberg adjusted *p*-value was calculated to control the false discovery rate (FDR) in multiple tests. KEGG pathways were considered significantly enriched by satisfying a LDA score of 1.6 and FDR corrected *p*-value of 0.05.

## 3. Results

### 3.1. Microbial Diversity

Principal coordinates analysis (PCoA) based on Bray-Curtis dissimilarity indicated that no significant difference was observed when samples were clustered, according to the sampling type, i.e., nasal and laryngeal. However, the structural pattern of the URT microbiota from pigs with clinical signs was clearly distinct from those without clinical signs at T1 (r^2^ = 0.61, *p* < 0.001). This difference was maintained at T2 (*p* < 0.001) and not observed in samples from pigs at the finishing phase (T3), which tended to cluster together, independent of the health status (Figure 1A,B). 

This result was confirmed by the pairwise PERMANOVA test (Table 1). 

When the results were further stratified, according to sampling time and health status, the significance increased (r^2^ = 0.73, *p* < 0.001), which indicates that these five categories (i.e., T1_symptomatic, T1_asymptomatic, T2_symptomatic, T2_asymptomatic, and T3, where all animals were asymptomatic) better explain the clustering of samples (Figure 1C,D; Table 2). 

Shannon and Simpson diversity indices indicated that asymptomatic pigs in T1 and T2 presented a higher diversity, in comparison to the symptomatic and asymptomatic pigs in T3 (*p* < 0.001) (Figure 2A,B). On the other hand, the number of observed OTUs was significantly lower only in symptomatic pigs from T1; a similar profile was observed for the Chao1 index (*p* < 0.001) (Figure 2C,D). 

No significant differences in alpha-diversity metrics were observed when samples were clustered according to their status (healthy and symptomatic) or anatomical site of sampling (nasal or laryngeal). 

### 3.2. Microbial Composition and Distribution

Given the structure of the microbial communities, microbial composition, and distribution analyses were based on the five identified sample groups (Figure 1; Table 2). Taxonomy-based analysis of bacterial communities identified 1345 bacterial taxa (OTUs), which belong to 318 genera, 117 families, and 19 phyla. Proteobacteria (53%) Firmicutes (35%), Bacteroidetes (8%), Actinobacteria (4%), and Tenericutes (1%) were the most prevalent phyla in all samples (Figure 3A and Figure 4A). These phyla were also differentially abundant among groups (Figure 5A). 

While Proteobacteria was abundant in T3, Firmicutes and Bacteroidetes were less frequent in this group. Actinobacteria was prevalent in asymptomatic pigs (asymptomatic-T1 and asymptomatic-T2), and Tenericutes were prevalent in asymptomatic-T1.

The most frequent families in all samples were Moraxellaceae (33%), Pasteurellaceae (17%), Streptococcaceae (16%), Lachnospiraceae (4%), and Lactobacillaceae (4%). The 10 most abundant families represented 91% of all observed taxa. Their composition and distribution in the samples are presented in Figure 3B and Figure 4B, respectively. In total, 19 families were differentially abundant among sample groups (Figure 5B).

The most frequent genera in all samples were *Moraxella* (32%), *Streptococcus* (16%), *Actinobacillus* (9%), *Pasteurellaceae_unclassified* (8%), and *Lactobacillus* (4%). The 10 most abundant genera represented 80% of all observed taxa. Their composition and distribution in the samples are presented in Figure 3C and Figure 4C, respectively. In total, 34 genera were differentially abundant among the analyzed groups (Figure 5B).

### 3.3. Core and Rare Microbiota

In order to characterize the core and rare microbiota of asymptomatic pigs and those with respiratory clinical signs, we further analyzed the frequency of abundant, rare, and unique OTUs in the samples. In total, 524 OTUs were shared among the five groups and may be considered the core microbiota, while 101 OTUs were exclusive to one group and considered the rare taxa (Figure 6A). The highest number of unique OTUs was observed in T3 (132 OTUs), followed by the T1 samples, while only two unique OTUs were observed in the T2 samples. From the 59 OTUs present in all samples, 8 had relative abundance >1%, and 51 had relative abundance between 1–0.01%. The highest number of shared OTUs in all samples belonged to Lachnospiraceae (12 OTUs), Prevotellaceae (11 OTUs), Ruminococcaceae (9 OTUs), and Moraxellaceae (4 OTUs); other relevant taxa include Clostridium_sensu_stricto_1 (2 OTUs), *Streptococcus* (2 OTUs), *Rothia* (1 OTU), and *Mycoplasma* (1 OTU). Between healthy and symptomatic pigs (from T1 and T2), 235 OTUs were shared, while 23 were unique to asymptomatic pigs and 41 were unique to symptomatic pigs (Figure 6B).

### 3.4. Functional Prediction

Metabolic prediction indicated that the clustering of samples was similar to the taxonomic analysis, with asymptomatic, symptomatic, and T3 in different clusters (Figure 7A); the only difference was that T3 was closely related to symptomatic pigs. 

In total, 28 differentially abundant pathways were identified among groups (Figure 7B), mainly related to carbohydrate, energy, amino acid, and nucleotide metabolism in symptomatic pigs (especially in T2). Other relevant pathways associated with symptomatic pigs in T2 included the metabolism of cofactors and vitamins (folate and other terpenoid-quinone biosynthesis), membrane transport systems (phosphotransferase system and ABC transporters), and biofilm formation. Only six pathways were related to healthy pigs or T3: ribosome and aminoacyl-tRNA biosynthesis (translation), oxidative phosphorylation (energy metabolism), and flagellar assembly (cellular motility), as well as starch/sucrose and galactose metabolism (carbohydrate metabolism).

## 4. Discussion

In this study, a prospective analysis was performed to compare the URT microbiota of asymptomatic and respiratory clinically-affected pigs at three critical stages of production, raised under the same management conditions. The results showed a different microbiota in symptomatic animals, when compared to asymptomatic, at the onset of the symptoms (T1), and the difference was maintained at the end of the nursery (T2), suggesting that respiratory clinical signs and/or the use of antibiotics to treat these animals had a long-term impact on URT microbiota. A previous study also observed an association between health status and richness/diversity of bacteria in weaned pigs, revealing a high frequency of genera such as *Moraxella* and *Weeksella*, which might act as potential probiotics [11]. On the other hand, bacteria commonly associated with pneumonia, such as *Mycoplasma hyopneumoniae*, *Pasteurella multocida*, *Streptococcus* sp., and *Actinobacillus pleuropneumoniae*, were more frequently detected in pigs with respiratory clinical signs than pigs with no signs, suggesting their involvement in respiratory disease [24]. 

It is important to point out that the pigs included in this study received five different antibiotics in their feed during the entire nursery phase; at the finishing phase, they received four antimicrobials in 80.3%, representing only 23 days without medication (Appendix A). The extensive use of antibiotics during the nursery and finishing phases is a common practice in pig production, not only to prevent disease in intensive production systems, which provide optimal conditions for the spread of pathogens. The impact of antibiotics on the swine nasal microbiota is a topic of major concern and has been already investigated in a few studies [25,26]. Correa-Fiz et al. (2019) observed that the removal of antimicrobial treatment early in piglets’ lives increased bacterial diversity at weaning and the abundance of favorable bacterial genera, such as *Prevotella* and *Lactobacillus* [25]. Moreover, changes in nasal microbiota composition improved the performance and health status of those piglets in the nursery phase. Zeineldin et al. (2018) showed a pronounced, antimicrobial-dependent microbial shift in the composition of nasal microbiota over time, showing that parenteral antimicrobial administration has a considerable impact on modulating the nasal microbiota structure [26]. In our study, the use of antibiotics may have altered the upper respiratory tract microbiota; however, a noticeable tendency to uniformization of the colonizing bacterial populations was identified at the finishing phase, when no significant differences were observed in the bacterial communities detected in pigs, with or without respiratory clinical signs. In addition to major changes in the microbiota, the main concern with the extensive use of antibiotics is the emergence of antimicrobial resistance, which has already been extensively demonstrated [27,28]. For these reasons, the development of novel non-antibiotic strategies to prevent infection in food-producing animals, subsequently increasing animal productivity, has been a topic of intense research in the swine industry [29,30].

The results related to alpha diversity revealed that, in T1, the number of observed OTUs was significantly lower in symptomatic pigs with a similar profile for Chao1 index. Since Chao1 is a metric based on the rare taxa and Shannon/Simpson considers the whole microbial community, these results suggest that the diversity of members of low abundancy of the community is primarily affected right before the signs of respiratory disease appear after weaning (T1), while the whole diversity community tends to get even at the end of the finishing period (T3). Different results were observed in a previous study, in which the oropharynx microbiota of pigs with respiratory disease had higher diversity [31].

Microbial communities detected in the nasal cavity and larynx were not significantly different. While the nasal microbial community is being extensively studied in animals and humans, the larynx is less explored. However, given the importance of both sites to respiratory diseases, a comparison between the microbial populations in these two anatomical sites was considered of interest. Correa-Fiz et al. (2016) observed a maximum of 1603 OTUs clustered in five phyla in the nasal cavity of pigs [11]. On the other hand, the larynx leads to the lower respiratory tract, and it is not exposed, as the nasal cavity is, which could limit the diversity of bacteria. This was not the case in the current study, since the bacterial populations in the nasal cavity and larynx were quite similar. 

The shared and unique OTUs also presented some interesting features; the number of unique OTUs decreased in T2 groups (with and without clinical signs), while it increased significantly in T3, indicating how distinctive the microbiome is at the finishing phase. When considering the health status, symptomatic pigs presented nearly twice the number of unique OTUs, when compared to asymptomatic pigs, indicating that the initial phase of respiratory disease is accompanied by a significant increase in the number of opportunistic microbial taxa.

Besides the differences associated with the clinical conditions of the piglets, significant differences in the structure and diversity of the microbiome were also related to the time of sampling (i.e., T1, T2, and T3), indicating that age and/or dietary changes had a major influence on the composition of the bacterial community. Although changes in the long-term development of the pig microbiome are not a novelty, our results have some differences, when compared to the consensus literature. According to recent studies, the diversity of the microbiome of healthy pigs feeding in a uniform condition and without the use of antibiotics increases with age [32,33]. On the contrary, our results showed that the diversity decreased in T3, which could be related to the cumulative effect of the extensive use of antibiotics in feed. Recent studies demonstrate that early-life microbial colonization is the most critical time for shaping intestinal and immune development, with perturbations to the microbiota during this time having long-lasting negative implications for the host [34,35]. Under the rearing conditions adopted at the farms sampled here, animals presented different microbiomes at an early age (T1 and T2), which adapted to a more uniform microbial community at finishing. We cannot state, however, that such adaptations were positive or negative for the animals; clearly, all pigs were able to reach the finishing phase in apparently healthy condition and good corporal state, suggesting that the animals were positively adapted to that particular rearing environment. 

A closer exploration of the overall changes in the nasal microbial membership among the groups revealed significant changes in the relative abundances of specific taxa, from phyla to genera, which may have contributed to the clinical signs of respiratory disease. At the genus level, *Actinobacillus*, *Streptococcus Porphyromonas, Veillonella,* and an unclassified genus of Pasteurellaceae were more abundant in pigs with clinical signs of respiratory disease. Similarly, other studies also detected a higher abundance of these genera in symptomatic pigs, suggesting their involvement in respiratory clinical signs [24,31]. The lack of differences in the microbiomes of animals at T3 (which displayed no clinical signs) and reduction of those differentially abundant taxa in T1 and T2 indicate that the dysbiosis related to respiratory signs was “resolved” at the finishing phase.

The genera *Actinobacillus* and *Pasteurella* compose the complex and diverse microbiota of the upper respiratory tract of pigs, but their species *Actinobacillus pleuropneumoniae* and *Pasteurella multocida* are also involved in the porcine respiratory disease complex, as primary or secondary pathogens [36]. The *A. pleuropneumoniae* maternal immunity decreases around 2–12 weeks of age, predisposing pigs to field infection [37]. Additionally, *P. multocida* is an opportunistic pathogen that is commonly associated with primary infections by viruses or other bacteria [38].

The predominance of the genus *Streptococcus* in symptomatic pigs in T1 and T2 may reflect a decrease in maternal antibodies against *S. suis*, which occurs at around 5–10 weeks [39]. This bacterium is commensal to the upper respiratory tract of pigs, but the association of stressful situations (e.g., weaning) with a low immunity level may increase its potential to invade and disseminate in the host, thus causing a systemic disease [40]. It is one of the most important bacterial swine pathogens affecting post-weaned pigs [41]. The infection caused by the pathogen not only results in severe economic losses but also raises animal welfare concerns. Not just this species, but also the genus *Streptococcus* is considered a potential reservoir for antibiotic resistance and represents a high risk of transmission of such resistance to other veterinary and human pathogens, due to the presence of mobile genetic elements carrying resistance genes transferable at high frequency. This issue highlights the need for further studies to evaluate the pathogenicity of *Streptococcus* strains in pigs with clinical signs of respiratory disease. In addition, whole metagenomic studies would allow for new insight into the diversity and dynamics of the antibiotic resistance genes that are harbored by the URT microbiota.

The relative abundance of *Porphyromonas* and *Veillonella* was also differentially abundant in pigs with clinical signs of respiratory disease. Although both genera are described as members of the core nasal microbiota, their relationship with respiratory diseases has not yet been clarified. For this reason, an investigation into the potential role of these neglected genera, in relation to the increased risk of developing respiratory diseases, should be the focus of further work.

An interesting result was the high presence of *Mycoplasma* sp. in both T1 groups (symptomatic and asymptomatic), followed by low abundance in T2 and T3. This genus is highly prevalent in Brazilian pig herds as a primary agent of pneumonia (*M. hyopneumoniae)*. It is possible that the large use of antibiotics active against mycoplasmas (six out of seven drugs) in our study contributed to the reduction of the bacterial load in the upper respiratory tract, especially in T2 and T3. It is already reported that the antibiotic treatment reduces, but does not eliminate, mycoplasmas from the respiratory tract [42].

The genera *Acinetobacter*, *Agathobacter*, *Anaerovibrio*, *Blautia*, *Catenibacterium*, Lachnospiraceae_unclassified, *Phascolarctobacterium*, *Prevotella_2*, *Prevotella_9*, *Rothia*, and *Subdoligranulum* were more abundant in both groups of asymptomatic pigs and could be associated with a healthy status. Indeed, some of these bacteria are considered healthy farm- and model-associated taxa, such as *Blautia, Prevotella, Rothia*, and Lachnospiraceae [12,31,43]. Although the *Acinetobacter* species is important in the emergence and spread of antimicrobial resistance, the prevalence of this genus in asymptomatic pigs may be an indicator of its beneficial impact on the maintenance of a healthy physiological conditions in pigs. *Lactobacillus* and *Moraxella* were more prevalent in asymptomatic pigs and T3 and could also be associated with a healthy status. The statistically significant predominance of these genera suggests a potential protective role against respiratory diseases, and further studies accessing the administration of targeted prebiotics and probiotics to maintain stability in the nasal microbiota might be a useful strategy to prevent or treat respiratory diseases.

To further explore the underlying changes observed in the microbiota, a functional prediction analysis was performed. While the starch/sucrose/galactose metabolism and prokaryotic oxidative phosphorylation pathway were significantly decreased in symptomatic pigs, the methane and sulfur pathways were increased, indicating a tendency for an anaerobic energy supply mechanism in the development of respiratory diseases. We hypothesize that during the initial phase of respiratory diseases, opportunistic pathogens adapt to those conditions, thus forming biofilms and changing to anaerobic metabolism, as well as increasing their capacity to metabolize carbohydrates, amino acids, and nucleotides. Indeed, anaerobes are relatively frequent pathogens in respiratory infections, and they are associated with a variety of diseases [44]. However, the role and molecular mechanisms of anaerobic bacteria and their metabolism in respiratory disease are not well-understood. The increase in membrane transport system pathways may also be an indicator of a pathogenic status. The ABC transport system, for example, is responsible for supplying energy for the uptake of a variety of nutrients and extrusion of drugs, such as antibiotics [45]. In addition, different studies indicate that these transporters play a significant role in the survival and proliferation of pathogens within the host [46]. Altogether, the results of the functional prediction indicate that identifying the metabolic pathways in symptomatic and asymptomatic pigs might be useful for the development of health-promoting strategies, based on the control of the metabolic activity of the microbiota in production systems.

## 5. Conclusions

The findings reported here provide evidence that the composition of the URT bacterial microbiota differs significantly when comparing pigs with or without respiratory clinical signs after weaning; these differences were maintained in the nursery phase but were not observed at the finishing phase. At the genus level, *Actinobacillus*, *Streptococcus Porphyromonas, Veillonella,* and an unclassified genus of Pasteurellaceae were more abundant in pigs with clinical signs of respiratory disease. Based on the observed changes in the metabolic pathways, we hypothesize that, during the initial phase of respiratory disease, opportunistic pathogens adapt to those conditions, thus forming biofilms and changing to anaerobic metabolism, as well as increasing their capacity to metabolize carbohydrates, amino acids, and nucleotides. Our results contribute to the knowledge of the porcine URT microbiota at different stages of production, providing new insights into the role of bacteria in the early stages of respiratory diseases.

## Figures and Tables

**Figure 1 biology-11-01111-f001:**
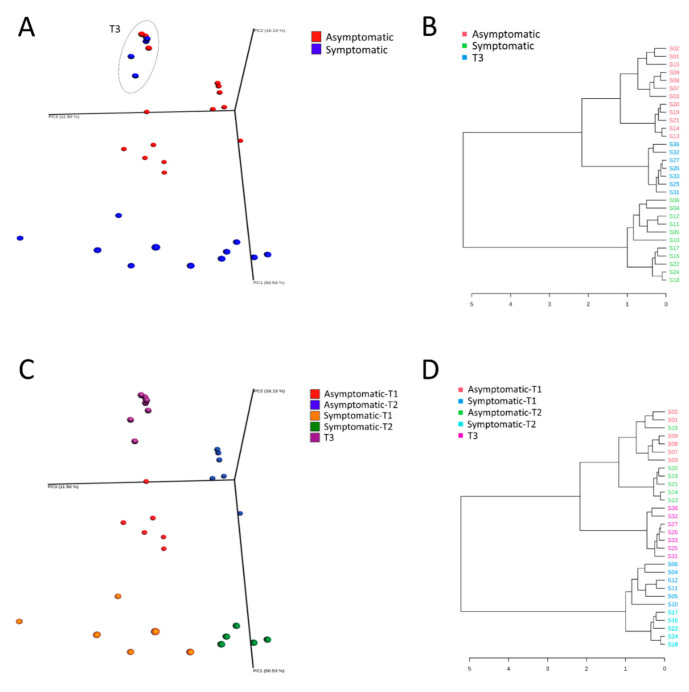
Beta diversity analysis based on Bray-Curtis distances in Asymptomatic and Symptomatic (with clinical symptoms of respiratory disease) pigs. Principal coordinates analysis (**A**) and dendrogram clustering (**B**) of the bacterial community structures grouped according to the pigs’ health status. Principal coordinates analysis (**C**) and dendrogram clustering (**D**) of the bacterial community structures grouped according to pigs’ health status and time of sampling. Statistical confidence for the sample grouping was accessed using permutational multivariate analysis of variance (PERMANOVA).

**Figure 2 biology-11-01111-f002:**
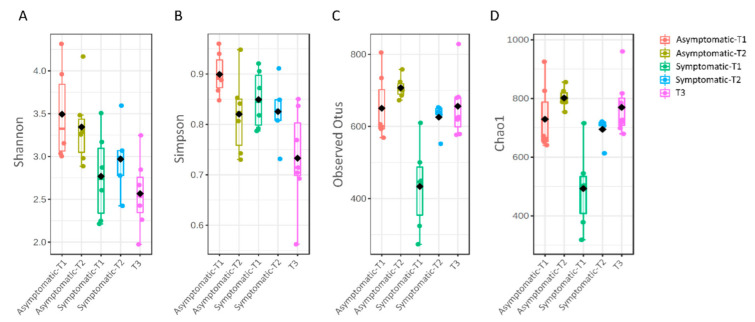
Alpha diversity analysis according to different metrics in Asymptomatic and Symptomatic (with clinical symptoms of respiratory disease) pigs. Shannon (**A**), Simpson (**B**), Chao1 (**C**), and Observed OTUs (**D**). Statistical confidence for the sample grouping was accessed using permutational multivariate analysis of variance (PERMANOVA).

**Figure 3 biology-11-01111-f003:**
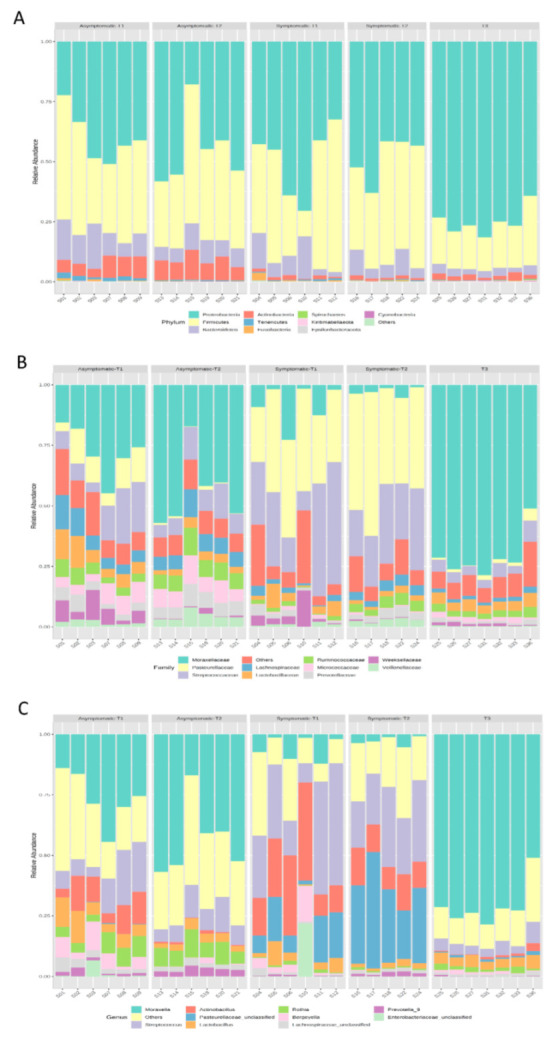
Relative abundance of the main phyla (**A**), families (**B**), and genera (**C**) observed in Asymptomatic and Symptomatic (with clinical symptoms of respiratory disease) pigs.

**Figure 4 biology-11-01111-f004:**
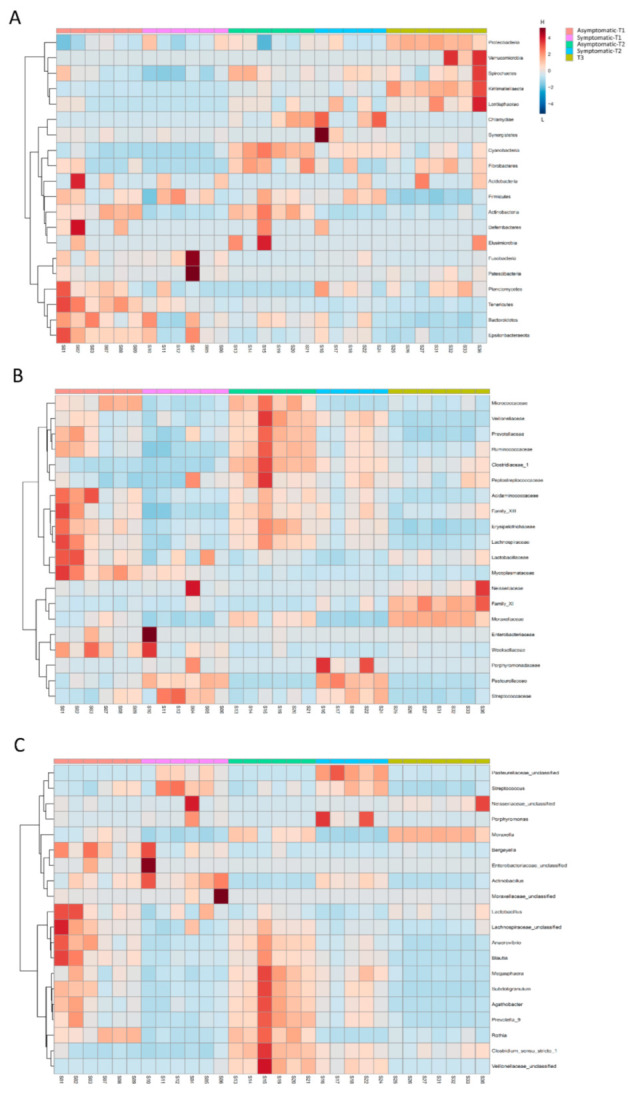
Distribution of the main phyla (**A**), families (**B**), and genera (**C**) observed in Asymptomatic and Symptomatic (with clinical symptoms of respiratory disease) pigs. The variation of taxonomic abundance was turned into a linear scale, from −4 (low abundance, L) to +4 (high abundance, H).

**Figure 5 biology-11-01111-f005:**
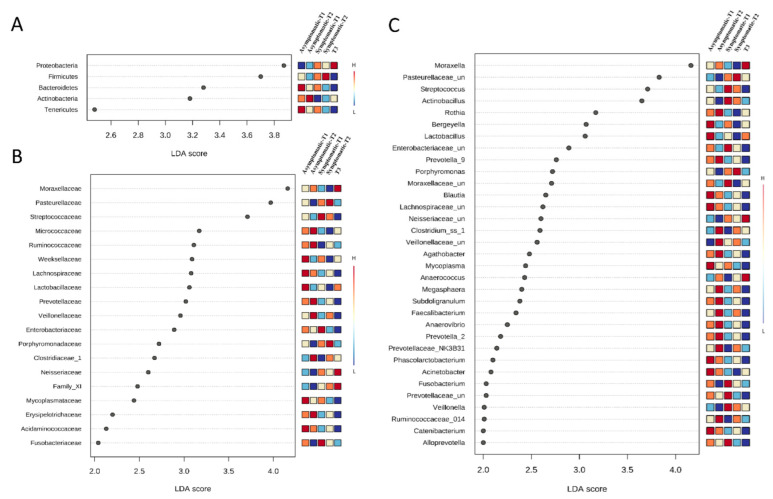
Lefse analysis at the phyla (**A**), family (**B**), and genus level (**C**) in Asymptomatic and Symptomatic (with clinical symptoms of respiratory disease) pigs. Statistical confidence was assessed using the Kruskal-Wallis test. Genera were ranked by LDA score. Differences were considered significant beyond a logarithmic LDA score threshold of ±2.0 and a *p*-value < 0.05.

**Figure 6 biology-11-01111-f006:**
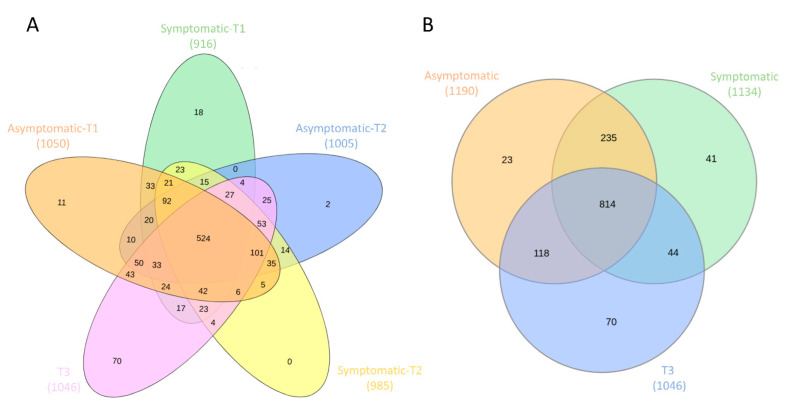
Venn diagram showing unique and shared operational taxonomic units (OTUs) among the 5 groups (**A**) and 3 groups (**B**) of samples observed in this study.

**Figure 7 biology-11-01111-f007:**
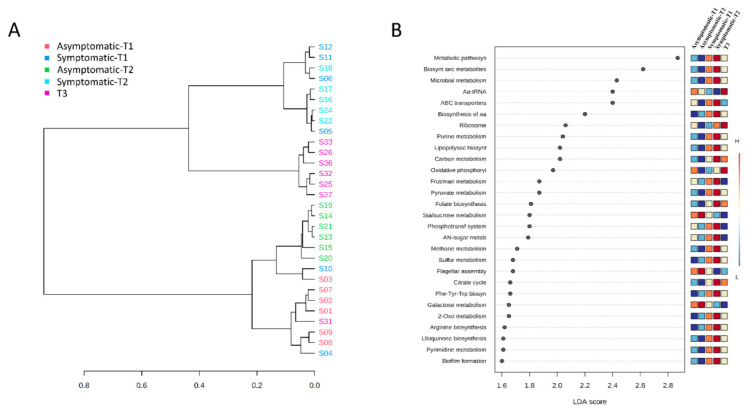
Functional prediction of the microbiome. Dendrogram clustering of samples according to the KEGG pathway identified (**A**). Linear discriminative analysis effect size (LEfSe) of statistically significant KEGG pathways among the 5 groups (**B**). Statistical confidence was assessed using the Kruskal-Wallis test. KEGG pathways were ranked by LDA score. Differences were considered significant beyond a logarithmic LDA score threshold of ±1.6 and a *p*-value < 0.05.

**Table 1 biology-11-01111-t001:** Pairwise PERMANOVA test among the group of samples identified in the beta diversity analysis based on Bray-Curtis distances in Asymptomatic and Symptomatic (with clinical symptoms of respiratory disease) pigs, and T3. The resulting *p*-values were corrected for multiple comparisons using the Benjamini–Hochberg correction (false discovery rate (FDR), *q*-value. A *q*-value < 0.05 was considered statistically significant.

Group 1	Group 2	Sample Size	*q*-Value
Symptomatic	Asymptomatic	23	0.001
Symptomatic	T3	18	0.001
Asymptomatic	T3	19	0.001

**Table 2 biology-11-01111-t002:** Pairwise PERMANOVA among the 5 groups of samples identified in the beta diversity analysis based on Bray-Curtis distances in Asymptomatic-T1, Symptomatic-T1, Asymptomatic-T2, Symptomatic-T2, and T3. The resulting *p*-values were corrected for multiple comparisons using the Benjamini–Hochberg correction false discovery rate (FDR), *q*-value. A *q*-value < 0.05 was considered statistically significant.

Group 1	Group 2	Sample Size	*q*-Value
Symptomatic-T1	Symptomatic-T2	11	0.01
Symptomatic-T1	Symptomatic-T3	10	0.01
Symptomatic-T1	Asymptomatic-T1	12	0.01
Symptomatic-T1	Asymptomatic-T2	12	0.01
Symptomatic-T1	Asymptomatic-T3	9	0.01
Symptomatic-T2	Symptomatic-T3	9	0.01
Symptomatic-T2	Asymptomatic-T1	11	0.01
Symptomatic-T2	Asymptomatic-T2	11	0.01
Symptomatic-T2	Asymptomatic-T3	8	0.01
Symptomatic-T3	Asymptomatic-T1	10	0.01
Symptomatic-T3	Asymptomatic-T2	10	0.01
Symptomatic-T3	Asymptomatic-T3	7	0.89
Asymptomatic-T1	Asymptomatic-T2	12	0.01
Asymptomatic-T1	Asymptomatic-T3	9	0.01
Asymptomatic-T2	Asymptomatic-T3	9	0.02

## Data Availability

The raw data of high-throughput sequencing is available at the NCBI GenBank Sequence Read Archive (SRA), under accession number PRJNA850682.

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
