# Peer review of "Comparative Analysis of the Upper Respiratory Bacterial Communities of Pigs with or without Respiratory Clinical Signs: From Weaning to Finishing Phase"

_biology, 2022, doi:10.3390/biology11081111_

Round 1

Reviewer 1 Report

The paper offers an interesting study analysis of the upper respiratory bacterial communities of pigs with or without respiratory clinical signs: from weaning to finishing phase. The authors, highlighted clearly, the difference between the different growth stage and the impact of antibiotic therapy on URT microbiota. Overall the paper was clear and well argued, discussions are a real strength of the paper in which researchers can find interesting food for thought. I propose it for publication in the present form.

Author Response

We thank the reviewer for reviewing and endorsing the paper.

Reviewer 2 Report

The manuscript “Comparative Analysis of the Upper Respiratory Bacterial Communities of Pigs With or Without Respiratory Clinical Signs: From Weaning to Finishing Phase” is generally well written but there are some aspects of the manuscript less clear that could benefit from some changes to increase its readability and interpretation:

  • The Results section starts with a description of the microbial diversity. As a suggestion, the authors could add a subsection related to the characterization of the sequencing data as this kind of information is not present in the article;
  • It would be more easy to interpret the section 2.1 Microbial Diversity if it appears after the section 2.2 Microbial Composition and distribution. An inversion in the order could make sense;
  • Line 378-381: The authors state “Antibiotic treatment in feed was administered as part of the farm management (Supplementary Table 1). Tulathromycin was administered intramuscularly … and Norfloxacin … was supplied in the drinking water when pigs presented respiratory clinical signs.”. Please clarify the following: the two groups of animals, symptomatic and asymptomatic pigs, were all treated with the same antibiotics except for Norfloxacin AND Tulathromycin which were used only in symptomatic pigs? It is not clear how the pigs were grouped because symptoms can appear between 33 and 173 days…

·       Since two of the antibiotics were administered only to symptomatic pigs, wouldn’t make sense the use of a control group of asymptomatic pigs also having the same antibiotics? Otherwise, the comparative results might be too speculative as no real control is used – i.e. there is more than one variable changing.

·       Not able to review Supplementary Materials (not available)

Author Response

Please find attached our reply.

Author Response

Please find attached our reply.
